# Effects of L-Theanine Administration on Stress-Related Symptoms and Cognitive Functions in Healthy Adults: A Randomized Controlled Trial

**DOI:** 10.3390/nu11102362

**Published:** 2019-10-03

**Authors:** Shinsuke Hidese, Shintaro Ogawa, Miho Ota, Ikki Ishida, Zenta Yasukawa, Makoto Ozeki, Hiroshi Kunugi

**Affiliations:** 1Department of Mental Disorder Research, National Institute of Neuroscience, National Center of Neurology and Psychiatry, 4-1-1, Ogawa-Higashi, Kodaira, Tokyo 187-8502, Japan; shidese@ncnp.go.jp (S.H.); sogawa@ncnp.go.jp (S.O.); ota@ncnp.go.jp (M.O.); iishida@ncnp.go.jp (I.I.); 2Department of Research and Development, Nutrition Division, Taiyo Kagaku Co. Ltd, 1-3, Takara-machi, Yokkaichi, Mie 510-0844, Japan; zyasukawa@taiyokagaku.co.jp (Z.Y.); mozeki@taiyokagaku.co.jp (M.O.)

**Keywords:** cognition, emotion, L-theanine, sleep, stress

## Abstract

This randomized, placebo-controlled, crossover, and double-blind trial aimed to examine the possible effects of four weeks L-theanine administration on stress-related symptoms and cognitive functions in healthy adults. Participants were 30 individuals (nine men and 21 women; age: 48.3 ± 11.9 years) who had no major psychiatric illness. L-theanine (200 mg/day) or placebo tablets were randomly and blindly assigned for four-week administration. For stress-related symptoms, Self-rating Depression Scale, State-Trait Anxiety Inventory-trait, and Pittsburgh Sleep Quality Index (PSQI) scores decreased after L-theanine administration (*p* = 0.019, 0.006, and 0.013, respectively). The PSQI subscale scores for sleep latency, sleep disturbance, and use of sleep medication reduced after L-theanine administration, compared to the placebo administration (all *p* < 0.05). For cognitive functions, verbal fluency and executive function scores improved after L-theanine administration (*p* = 0.001 and 0.031, respectively). Stratified analyses revealed that scores for verbal fluency (*p* = 0.002), especially letter fluency (*p* = 0.002), increased after L-theanine administration, compared to the placebo administration, in individuals who were sub-grouped into the lower half by the median split based on the mean pretreatment scores. Our findings suggest that L-theanine has the potential to promote mental health in the general population with stress-related ailments and cognitive impairments.

## 1. Introduction

L-theanine (γ-glutamylethylamide) is a unique non-protein amino acid found in green tea (Camellia sinensis), a widely consumed beverage associated with human health [1,2,3]. As the structure of L-theanine resembles that of L-glutamic acid, its mechanism of action may be potentially mediated through glutamate receptors [4], a possibility supported by its partial co-agonistic effect on the N-methyl-D-aspartate receptor [5,6]. Given that L-theanine is a phytochemical ingested in daily life, it has the potential to become a nutraceutical ingredient that mitigates and prevents stress-related psychic confusion in modern society [7,8,9,10].

In rodents, L-theanine has been proposed as a neuroprotective and cognitive-enhancing agent [11]. Peripherally and centrally administered L-theanine has been reported to act on the brain [12,13], to modulate monoamine levels in the central nervous system [14,15,16,17], and glutamate and glycine neurotransmissions [18,19]. L-theanine has been also reported to influence hippocampal synaptic plasticity, improving performance in recognition tasks in stressed young rats [20,21]. Furthermore, in behavioral experiments in mice, L-theanine has been reported to have anti-stress or antidepressant-like effects [22,23,24,25,26], which has been associated with an increase in the expression of brain-derived neurotrophic factor in the hippocampus [6], while anxiolytic effects of L-theanine have been observed in Wistar Kyoto rats, where it enhances hippocampal activity in positron emission tomography [27].

In humans, L-theanine has been reported to modulate alpha activity and play a role in attentional tasks in electroencephalogram studies [28,29,30], as well as provide beneficial effects on mental state [31], including sleep quality [32]. To our knowledge, however, only its acute effects have been investigated in healthy individuals. The anti-stress effects of L-theanine (200 mg/day) have been observed following once- [33,34] and twice daily [35] administration, while its attention-improving effects have been observed in response to treatment of 100 mg/day on four separate days [36] and 200 mg/day single administration [37], which was further supported by decreased responses in functional magnetic resonance imaging [38]. Moreover, L-theanine has been suggested to have potential therapeutic effects in psychiatric disorders [39]. In accordance with this, we have reported multiple favourable effects of four weeks L-theanine administration (250 mg/day) in patients with major depressive disorder (MDD), although this was an open-label study [40].

With the except of Hidese et al. [40], most human studies have focused on the effects of acute L-theanine administration. The present study aimed to examine the effects of four weeks L-theanine administration (200 mg/day, four weeks) in a healthy population, i.e., individuals without any major psychiatric disorder. Namely, the population of this study comprised individuals suffering from some non-clinical symptoms (e.g., emotion, sleep, and cognition). In order to obtain valid data, this study was designed to be a randomized, placebo-controlled, crossover, and double-blind trial.

## 2. Materials and Methods 

### 2.1. Participants

Thirty individuals (Nine men and 21 women; mean age: 48.3 ± 11.9 years [range: 20–69]), without any major psychiatric illness and enrolled after a screening by a trained psychiatrist, participated in the study. The sample size was determined by using G*Power software (3.1 version, Franz Faul, University of Kiel, Kiel, Germany) [41,42]. The current sample (30 L-theanine and 30 placebo) had a power of> 0.80, based on the effect size (0.61) of L-theanine on depression score in our previous study [40]. Individuals who had been treated for any psychiatric disorder, assessed according to the criteria of the Japanese version of the Mini-International Neuropsychiatric Interview [43,44] and the Diagnostic and Statistical Manual of Mental Disorders, 5th edition [45], were excluded. Individuals who did not have any psychiatric disorders were included, while those who had severe physical illnesses or pregnant women were excluded. We obtained the information about current drinking alcohol and smoking habits. All participants were recruited randomly through prepared advertisements in the local community and announcements in a free magazine or on our website. Written informed consent was obtained from every participant. Confidentiality of personal information was consistently preserved by anonymizing data obtained by using code corresponding to personal identification information. The protocol was approved by the ethics committee at the National Center of Neurology and Psychiatry, and primary and secondary endpoints were complied with, accordingly. The study protocol was registered at the university hospital medical information network—clinical trials registry (UMIN 000028603) and carried out according to the Declaration of Helsinki [46]. This randomized study adhered to the Consolidated Standards of Reporting Trials extensions [47]. 

### 2.2. L-theanine Administration

An enzymatic synthetic product L-theanine (200 mg/day; Suntheanine, Taiyo Kagaku Co., Ltd, Mie, Japan) or placebo tablets were orally self-administered before sleep each night for four weeks, while the compliance was checked only verbally. Placebo tablet was the same as L-theanine tablet in terms of appearance and taste. This dose was chosen because it was found to be effective in a previous study [48], which was normal dose as also used in other studies [33,34,35,37]. However, the dose was relatively high compared to the median amount of L-theanine per cup of green tea (8–30 mg) [49]. The tablets were allocated to the participants in a randomized manner, using a table of random numbers to generate the sequence for randomization, by a representative assignee who was not an intervention practitioner. After a two-week wash-out period, which was justified based on the half-life of L-theanine plasma concentration reported around 65 min in humans [50], participants received the other tablets for a further four weeks, according to the crossover method. Participants were instructed not to alter their usual intake of green tea or other beverage contained L-theanine during the intervention period, although it was not actually tracked. In accordance with double-blind procedures, the identity of the active or placebo tablets was not disclosed to participants or research conductors at the National Center of Neurology and Psychiatry until all the statistical analyses had been completed. The tablets were labeled only as either A or B at the time of supply to the participants. After performance of the statistical analyses by an outcome evaluator, the identity of the tablets was finally disclosed by a researcher at Taiyo Kagaku Co., Ltd. All the data collection and analysis were thus performed by blinded investigators. After removing anonymity, L-theanine and placebo were found to have been administered to 16 and 14 participants, respectively, between weeks zero and four. Following the two-week wash-out period, subjects were administered tablets in a crossover manner between weeks six and ten.

### 2.3. Clinical Assessments (Primary Outcome)

Clinical assessments were performed at baseline, and four, six, and ten weeks into the trial. The Japanese version of the Self-rating Depression Scale (SDS) [51,52], the State-Trait Anxiety Inventory (STAI) [53,54], and the Pittsburgh Sleep Quality Index (PSQI) [55,56] were used to evaluate depression, anxiety, and sleep quality, respectively. Cognitive functions were evaluated by a research psychiatrist using the Trail-Making Test [57], the Stroop test [58], and the Japanese version of the Brief Assessment of Cognition in Schizophrenia (BACS) [59,60]. The Stroop test was conducted on a computer using two separate PowerPoint files for control and experimental conditions, composed of four colored words (blue, red, yellow, and black). After practice with the control condition, wherein the color and word were matched, response latency (msec) and error rate (%) were calculated in the experimental condition, in which the colour and word were different. Two different versions (A and B) of the Stroop and BACS tests were used alternately at each assessment to exclude the possibility of any learning effect. All assessments completed on the same day from 10:00 to 12:00.

### 2.4. Laboratory Tests (Secondary Outcome)

Fasted venous blood and salivary samples were obtained between 12:00 and 13:00 after the clinical assessments at baseline and four, six, and ten weeks into the trial. Serum and plasma samples were prepared by centrifuging blood for ten minutes at 2000× *g*, while salivary supernatants were prepared by centrifuging saliva impregnated in mesh (Salimetrics, LLC., Carlsbad, California, United States of America) for 15 min at 1000× *g* (Kubota Corp., Tabletop Centrifuge Model 4000, Tokyo, Japan). In addition to general serum measures, plasma glucose and haemoglobin-A1c, and whole blood count levels, serum cortisol and immunoglobulin-A levels were measured at SRL Co., Ltd. (Tokyo, Japan). Salivary cortisol and immunoglobulin-A levels were measured at our laboratory using Salivary enzyme immunoassay kits (1-3002 and 1-1602; respectively, Salimetrics, LLC., Carlsbad, California, United States of America). Serum L-theanine concentrations were measured using a high-performance liquid chromatography system based on the AccQ·Tag method [61], using commercially available reagents (Waters Corp., Tokyo, Japan) and equipment (Shimazu Corp., Kyoto, Japan) at Taiyo Kagaku Co., Ltd. (Mie, Japan) The anonymity of the participants was preserved throughout the biochemical tests.

### 2.5. Statistical Analyses

The crossover data during the intervention period were combined to a total period (i.e., the sum of baseline-four and six-ten weeks) of four weeks L-theanine and placebo administration. Clinical variables were compared between baseline and after four weeks of administration, using the Wilcoxon signed-rank test. Changes in clinical variables between pre- and post-treatment were compared between the L-theanine and placebo administrations, using the Mann-Whitney U test. The effect sizes were calculated using r values from the Wilcoxon signed-rank and Mann-Whitney U tests. For comparison of the BACS scores, we performed analyses stratified by the median split based on the mean pretreatment scores (i.e., dividing the 30 subjects into two groups [*N* = 15 each] based on the average of the scores at week zero and that of week six) for each cognitive function. Changes in BACS scores were compared between L-theanine and placebo administrations in the high and low score groups, separately. All statistical tests were two-tailed and *p* < 0.05 was deemed statistically significant. Statistical analyses were performed using the Statistical Package for the Social Sciences version 25.0 (IBM Corp., Tokyo, Japan).

## 3. Results

### 3.1. Participants

The clinical characteristics of the participants are shown in Table 1. The data collection period was from 7th June 2017 to 7th August 2018. There was no participant drop-out (i.e., trial compliance of 100%), and adverse events were not apparent during the intervention period.

### 3.2. Symptom Scores

The stress-related symptom scores after 4 weeks L-theanine or placebo administration are shown in Table 2. The SDS, STAI-trait (T), and PSQI scores were significantly improved after L-theanine administration (*p* = 0.019, 0.006, and 0.013, respectively; Figure 1). Among the PSQI subscales, the scores of “sleep latency” and “daytime dysfunction” significantly improved in response to L-theanine administration (*p* = 0.036 and 0.022, respectively). In contrast, there was no significant difference after placebo administration. When score reductions in the stress-related symptoms were compared between L-theanine and placebo administrations, changes in the PSQI sleep latency, sleep disturbance, and use of sleep medication subscales were significantly greater (*p* = 0.0499, 0.046, and 0.047, respectively), while those in the SDS and PSQI scores showed a non-statistically significant trend towards greater improvement (*p* = 0.084 and 0.073, respectively), during the L-theanine period compared to placebo.

### 3.3. Cognitive Function Scores

The cognitive function scores after 4 weeks L-theanine or placebo administration are shown in Table 3. The BACS verbal fluency, especially letter fluency (*p* = 0.001), and executive function scores were significantly increased after L-theanine administration (*p* = 0.001 and 0.031, respectively; Figure 2), while the Trail Making Test A and B scores were significantly improved after placebo administration (*p* = 0.042 and 0.038, respectively). When score changes were compared between the L-theanine and placebo administrations, no significant difference was found for any cognitive function. We then performed analyses stratified by the median split based on the mean pretreatment scores for each cognitive function (Table 4). We found that improvement in the BACS verbal fluency (*p* = 0.002), especially letter fluency (*p* = 0.002), score was significantly greater during the L-theanine period than the placebo period among those individuals who were sub-grouped into the lower half by the median split (*n* = 15). In addition, the BACS executive function score improvement tended to be greater in the L-theanine period than in the placebo period among the lower half of the group (*p* = 0.062). Furthermore, we repeated the analyses with subjects restricted to those aged 40 years or more, since cognitive decline typically begins at, or after, middle age. The results were essentially unchanged (*n* = 21; verbal fluency: *p* = 0.023, letter fluency: *p* = 0.040, and executive function: *p* = 0.048; Appendix A).

### 3.4. Biochemical Data

Body mass index and biochemical data after 4 weeks L-theanine or placebo administration are shown in Appendix A. The serum L-theanine concentration was significantly increased after L-theanine administration (*p* = 0.002), while it was not significantly altered after placebo administration. There was a significant difference in the serum L-theanine concentration between the L-theanine and placebo administrations (*p* = 0.028). No other significantly altered indices were found after either L-theanine or placebo administration.

## 4. Discussion

In this placebo-controlled study, stress-related symptoms assessed with SDS, STAI-T, and PSQI scores decreased, while BACS verbal fluency and executive function scores improved following four weeks L-theanine administration. Reductions in PSQI subscales were greater, and improvement in BACS verbal fluency, especially letter fluency, was higher during the L-theanine administration, compared to the placebo, among those who showed poorer baseline functions. These results suggest that four weeks L-theanine administration has positive effects on stress-related symptoms and cognitive function in a healthy population. To our knowledge, this is the first study that has examined stress-related symptoms and cognitive functions simultaneously in healthy adults.

For stress-related symptoms, SDS, STAI-T, and PSQI scores decreased after four weeks L-theanine administration. These findings are consistent with previous studies in rodents [6,22,23,24,25,26,27], acute effects reported in healthy humans [31,32,33,34,35], and chronic effects in patients with MDD [40]. Additionally, this trial suggests, for the first time, that 4 weeks L-theanine administration is effective in improving stress-related symptoms in healthy adults. The effects on stress-related symptoms were broad among the symptom indices presented in the study, although a comparison to the placebo administration somewhat limits the efficacy of L-theanine administration for some sleep disturbance measurements.

For cognitive functions, BACS verbal fluency and executive function scores improved after four weeks L-theanine administration. These findings are consistent with previous preclinical studies [11,20,21] and our clinical trial for MDD [40], whereas they are inconsistent with the acute attention-improving effects reported in healthy humans [28,29,30,36,37,38]. Considering the comparison to the placebo administration, the current study suggests that the score for the BACS verbal fluency, especially letter fluency, but not the Trail Making Test, Stroop test, or other BACS parameters, significantly changes in response to the 4 weeks effects of L-theanine. In contrast to our previous study [40], the present study was performed on individuals without clinical depression, which may have resulted in inconsistency concerning the cognitive-enhancing effects. This is partly consistent with the finding that the present verbal fluency enhancement occurred in the subgroup in which cognitive function was poorer at pretreatment.

There was no significant alteration in body mass index or biochemical data after four weeks L-theanine administration. It should also be noted that there were no significant adverse events, demonstrating the safety of four weeks of L-theanine administration. Although psychotropic effects were observed in the current study, four weeks L-theanine administration had no significant effect on cortisol or immunoglobulin A levels in the saliva or serum, which was inconsistent with previous studies reporting that salivary cortisol [34] and immunoglobulin A [33] levels were reduced after acute L-theanine administration. Considering this inconsistency, the reduction of salivary cortisol and immunoglobulin A levels may be short term, recoverable effect of L-theanine administration. As expected, changes in serum L-theanine concentrations were higher in the L-theanine administration compared to the placebo administration, which validated the compliance of the participants in this intervention.

L-theanine may act as a partial agonist for the N-methyl-D-aspartate receptor [5], resulting in its four weeks psychotropic (i.e., antidepressant, anxiolytic, sleep aid, and cognitive-enhancing) effects. The modulation of the monoamine [14,15,16,17] and glutamic acid and glycine [18,19] systems may also contribute to these four weeks psychotropic effects. Hippocampal function may be related to cognitive-enhancing effects as suggested in our prior studies [6,27]. In addition to the possible efficacy in the treatment of clinical depression [40], the present study suggests that L-theanine may be a psychoactive reagent broadly useful to improve mental health in humans. Moreover, our data support the translational value of L-theanine considering its multiple psychoactive effects which have been observed in animal studies.

There are several limitations to the study. First, this trial was conducted in a real-world setting and possible biases (e.g., green or other kind of tea consumption, medicine intake, diet, intake nutrients, and type of work) were not accounted for, which may rather impact the efficacy of L-theanine in daily life. Second, only about 20% of symptoms (the PSQI subscales) and cognitive functions (the BACS verbal fluency, especially letter fluency and executive function) scores showed significant changes after L- theanine administration compared to the placebo administration, suggesting that the effects are not large on daily function of the participants. Third, any psychotropic effects of L-theanine may have been relatively inconspicuous as our participants had no psychiatric disorders, whereas these effects may be more significant in severe clinical cases. In this context, the study was conducted on a small population and the number of the participants (*n* = 30) may have been insufficient, which is subject to type II errors, although the sample size had been determined according to the effect size from our previous study in patients with MDD [40]. Finally, one might suspect that a tighter age range and one sex population may have been more appropriate. However, we had no solid information on any specific age range or sex in which L-theanine is effective. We therefore included subjects of a relatively wider range of age and both sexes. Further studies are warranted to confirm the presented effects in much larger sample size.

## 5. Conclusions

Stress-related symptom (i.e., depression, anxiety-trait, and sleep) scores decreased and cognitive function (i.e., verbal fluency and executive function) scores improved after four weeks of L-theanine administration. The reduction in sleep quality problems (disturbances in sleep latency, sleep disturbance, and use of sleep medication) was greater in the L-theanine administration compared to the placebo administration, while verbal fluency, especially letter fluency, was improved in the L-theanine administration among individuals who showed relatively low performance at pretreatment. Moreover, L-theanine administration was safe and well complied with. Therefore, L-theanine may be a suitable nutraceutical ingredient for improving mental conditions in a healthy population.

## Figures and Tables

**Figure 1 nutrients-11-02362-f001:**
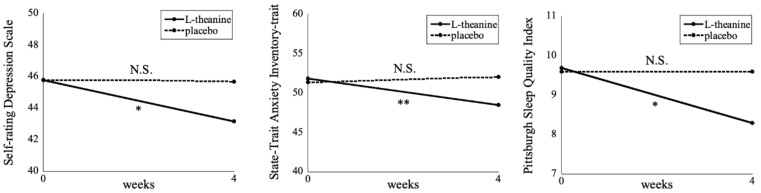
Stress-related symptom scores after 4 weeks L-theanine or placebo administration. The Wilcoxon signed-rank test revealed that the Self-rating Depression Scale, State-Trait Anxiety Inventory-trait, and Pittsburgh Sleep Quality Index scores were significantly decreased in the L-theanine administration (*n* = 30), while there was no significant change in the placebo administration (*n* = 30). * *p* < 0.05, ** *p* < 0.01. N.S., not significant.

**Figure 2 nutrients-11-02362-f002:**
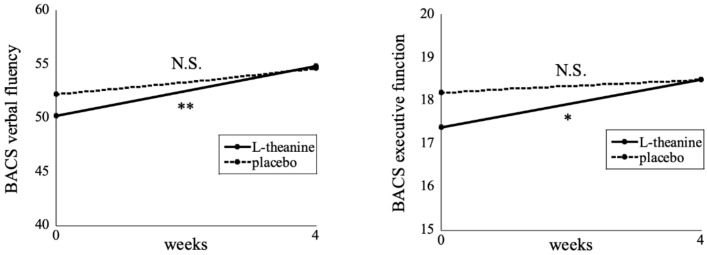
Cognitive function scores after 4 weeks L-theanine or placebo administration. The Wilcoxon signed-rank test revealed that the Brief Assessment of Cognition in Schizophrenia (BACS) verbal fluency and executive function scores were significantly decreased in the L-theanine administration (*n* = 30), while there was no significant change in the placebo administration (*n* = 30). * *p* < 0.05, ** *p* < 0.01. N.S.: Not significant.

**Table 1 nutrients-11-02362-t001:** The clinical characteristics of the participants.

Variables	Mean ± Standard Deviation	Range
Age (years)	48.3 ± 11.9	20–69
Sex, female: *n* (%)	21 (70)	
Education (years)	14.5 ± 2.0	10–18
Height (cm)	161 ± 8.0	148–180
Weight at baseline (kg)	58.6 ± 12.7	44–102
Body mass index at baseline (kg/m^2^)	22.5 ± 3.9	17.1–31.5
Drinking alcohol habit: *n* (%)	20 (66.7)	
Smoking habit: *n* (%)	8 (26.7)	

*n* = 30, frequency (percentage).

**Table 2 nutrients-11-02362-t002:** The stress-related symptom scores after 4 weeks L-theanine or placebo administration.

Symptoms	Pretreatment	Posttreatment	Wilcoxon Signed-Rank Test	Pretreatment	Posttreatment	Wilcoxon Signed-Rank Test	Score Changes		Mann-Whitney U Test
	L-theanine			Placebo			L-theanine	Placebo	
Self-rating Depression Scale	45.8 ± 7.38	43.2 ± 7.47	W = 83.5, ***p* = 0.019**, r = −0.43	45.8 ± 7.41	45.7 ± 6.77	W = 216.0, *p* = 0.77, r = −0.06	−2.53 ± 5.38	−0.07 ± 6.60	U = 333.5, *p* = 0.084, r = −0.22
State-Trait Anxiety Inventory-state	48.6 ± 9.32	46.1 ± 9.51	W = 158.5, *p* = 0.20, r = −0.23	50.6 ± 9.86	49.9 ± 11.2	W = 181.5, *p* = 0.44, r = −0.14	−2.47 ± 9.26	−0.67 ± 8.75	U = 435.0, *p* = 0.82, r = −0.03
State-Trait Anxiety Inventory-trait	51.9 ± 9.66	48.5 ± 10.4	W = 74.0, ***p* = 0.006**, r = −0.51	51.4 ± 10.3	52.1 ± 11.2	W = 184.0, *p* = 0.90, r = 0.02	−3.37 ± 8.13	0.77 ± 7.52	U = 348.0, *p* = 0.13, r = −0.20
Pittsburgh Sleep Quality Index	9.67 ± 2.71	8.3 ±2.45	W = 78.5, *p* = 0.013, r = −0.46	9.63 ± 2.24	9.60 ± 2.84	W = 132.0, *p* = 0.85, r = 0.03	−1.37 ± 2.81	−0.03 ± 2.08	U = 330.0, *p* = 0.073, r = −0.23
C1 (sleep quality)	1.77 ± 0.73	1.50 ± 0.73	W = 10.0, *p* = 0.052, r = −0.36	1.73 ± 0.69	1.63 ± 0.56	W = 30.0, *p* = 0.44, r = −0.14	−0.27 ± 0.74	−0.10 ± 0.71	U = 403.0, *p* = 0.42, r = −0.11
C2 (sleep latency)	1.67 ± 1.12	1.23 ± 1.01	W = 34.0, ***p* = 0.036**, r = −0.38	1.63 ± 0.93	1.63 ± 0.93	W = 27.5, *p* = 1.00, r = 0.00	−0.43 ± 1.04	0.00 ± 0.59	U = 330.0, ***p* = 0.0499**, r = −0.25
C3 (sleep duration)	3.00 ± 0.00	3.00 ± 0.00	W = 0.0, *p* = 1.00, r = not applicable	3.00 ± 0.00	3.00 ± 0.00	W = 0.0, *p* = 1.00, r = not applicable	0.00 ± 0.00	0.00 ± 0.00	U = 450.0, *p* = 1.00, r = 0.00
C4 (habitual sleep efficiency)	0.50 ± 0.94	0.50 ± 0.86	W = 30.0, *p* = 0.93, r = −0.02	0.63 ± 0.96	0.53 ± 0.90	W = 22.0, *p* = 0.57, r = −0.11	0.00 ± 0.98	−0.10 ± 0.92	U = 435.0, *p* = 0.79, r = 0.03
C5 (sleep disturbances)	1.27 ± 0.45	1.10 ± 0.40	W = 10.0, *p* = 0.096, r = −0.31	1.17 ± 0.59	1.27 ± 0.64	W = 20.0, *p* = 0.26, r = 0.21	−0.17 ± 0.53	0.10 ± 0.48	U = 345.5, ***p* = 0.046**, r = −0.26
C6 (use of sleeping medication)	0.27 ± 0.78	0.13 ± 0.57	W = 0.0, *p* = 0.18, r = −0.25	0.13 ± 0.57	0.27 ± 0.78	W = 3.0, *p* = 0.18, r = 0.25	−0.13 ± 0.57	0.13 ± 0.57	U = 392.0, ***p* = 0.047**, r = −0.26
C7 (daytime dysfunction)	1.20 ± 1.00	0.83 ± 0.79	W = 32.0, ***p* = 0.022**, r = −0.42	1.33 ± 0.84	1.27 ± 0.91	W = 115.0, *p* = 0.67, r = −0.08	−0.37 ± 0.81	−0.07 ± 0.87	U = 374.0, *p* = 0.23, r = −0.15

Values are represented as mean ± standard deviation. Significant *p*-values are shown in bold cases.

**Table 3 nutrients-11-02362-t003:** The cognitive function scores after 4 weeks L-theanine or placebo administration.

Cognitive functions	Pretreatmnent	Posttreatment	Wilcoxon Signed-Rank Test	Pretreatment	Postreatment	Wilcoxon Signed-Rank Test	Score Changes		Mann-Whitney U Test
	L-theanine			Placebo			L-theanine	Placebo	
Trail Making Test A (sec)	27.4 ± 9.78	25.9 ± 10.2	W = 170.5, *p* = 0.31, r = −0.19	29.1 ± 10.7	25.9 ± 10.3	W = 123.5, ***p* = 0.042**, r = −0.37	−1.57 ± 9.68	−3.27 ± 7.48	U = 410.0, *p* = 0.55, r = 0.08
Trail Making Test B (sec)	62.4 ± 22.0	59.6 ± 24.8	W = 136.5, *p* = 0.13, r = −0.28	63.9 ± 22.9	58.6 ± 20.4	W = 121.5, ***p* = 0.038**, r = −0.38	−2.83 ± 18.5	−5.23 ± 14.9	U = 428.0, *p* = 0.75, r = 0.04
Stroop test response latency (msec)	1176 ± 219	1159 ± 228	W = 149.0, *p* = 0.50, r = −0.12	1188 ± 259	1140 ± 217	W = 154.5, *p* = 0.27, r = −0.20	−16.7 ± 158	−47.9 ± 183	U = 422.5, *p* = 0.68, r = 0.05
Stroop test error rate (%)	1.04 ± 1.62	1.81 ± 2.37	W = 130.0, *p* = 0.15, r = 0.26	0.83 ± 1.51	1.32 ± 2.35	W = 74.0, *p* = 0.42, r = 0.15	0.76 ± 2.41	0.49 ± 2.10	U = 426.0, *p* = 0.71, r = 0.05
Brief Assessment of Cognition in Schizophrenia									
verbal memory	52.2 ± 7.68	52.5 ± 7.74	W = 183.0, *p* = 0.85, r = 0.04	52.7 ± 9.05	51.5 ± 9.69	W = 182.0, *p* = 0.63, r = −0.09	0.30 ± 8.57	−1.20 ± 9.53	U = 414.5, *p* = 0.60, r = 0.07
working memory	20.9 ± 4.14	21.5 ± 4.43	W = 194.5, *p* = 0.39, r = 0.16	21.5 ± 4.91	21.4 ± 4.83	W = 158.5, *p* = 0.91, r = −0.02	0.57 ± 3.87	−0.03 ± 3.31	U = 397.0, *p* = 0.43, r = 0.10
motor speed	83.7 ± 12.6	84.0 ± 12.5	W = 193.5, *p* = 0.65, r = 0.08	83.2 ± 13.0	83.6 ± 13.3	W = 204.5, *p* = 0.71, r = 0.07	0.37 ± 7.80	0.43 ± 7.94	U = 439.5, *p* = 0.88, r = −0.02
verbal fluency	50.3 ± 13.0	54.9 ± 10.4	W = 364.5, ***p* = 0.001**, r = 0.58	52.3 ± 9.80	54.7 ± 12.2	W = 299.5, *p* = 0.076, r = 0.33	4.57 ± 7.13	2.33 ± 7.26	U = 336.5, *p* = 0.22, r = 0.20
category fluency	21.9 ± 5.87	23.1 ± 5.30	W = 271.5, *p* = 0.12, r = 0.29	22.4 ± 4.48	23.3 ± 5.49	W = 244.0, *p* = 0.35, r = 0.17	1.20 ± 4.23	0.90 ± 5.91	U = 442.5, *p* = 0.91, r = 0.02
letter fluency	28.4 ± 8.43	31.7 ± 6.94	W = 348.5, ***p* = 0.001**, r = 0.61	29.9 ± 7.28	31.4 ± 9.68	W = 283.5, *p* = 0.29, r = 0.19	3.37 ± 4.63	1.43 ± 5.83	U = 344.0, *p* = 0.12, r = 0.20
attention	64.4 ± 11.6	62.9 ± 11.7	W = 139.0, *p* = 0.53, r = −0.12	62.6 ± 11.3	62.9 ± 10.1	W = 204.0, *p* = 0.47, r = 0.13	−1.47 ± 7.37	0.33 ± 4.97	U = 308.5, *p* = 0.30, r = −0.13
executive function	17.4 ± 2.71	18.5 ± 1.72	W = 208.5, ***p* = 0.031**, r = 0.40	18.2 ± 1.77	18.5 ± 1.63	W = 205.0, *p* = 0.24, r = 0.21	1.10 ± 2.52	0.33 ± 1.81	U = 397.0, *p* = 0.43, r = 0.10

Values are represented as mean ± standard deviation. Significant *p*-values are shown in bold cases.

**Table 4 nutrients-11-02362-t004:** Comparisons of BACS score changes between 4 weeks L-theanine and placebo administrations stratified by the median split based on mean pretreatment score.

	L-theanine	Placebo	Mann-Whitney U Test
Verbal memory	0.30 ± 8.57	−1.20 ± 9.53	U = 414.5, *p* = 0.60, r = 0.07
Upper half	−3.13 ± 9.36	−3.07 ± 10.0	U = 108.0, *p* = 0.85, r = −0.04
Lower half	3.73 ± 6.27	0.67 ± 8.93	U = 95.0, *p* = 0.47, r = 0.13
Working memory	0.57 ± 3.87	−0.03 ± 3.31	U = 397.0, *p* = 0.43, r = 0.10
Upper half	0.47 ± 3.50	−0.80 ± 3.38	U = 80.0, *p* = 0.17, r = 0.25
Lower half	0.67 ± 4.34	0.73 ± 3.15	U = 111.0, *p* = 0.95, r = −0.01
Motor speed	0.37 ± 7.80	0.43 ± 7.94	U = 439.5, *p* = 0.88, r = −0.02
Upper half	0.67 ± 7.10	0.27 ± 6.24	U = 104.0, *p* = 0.72, r = 0.07
Lower half	0.07 ± 8.69	0.60 ± 9.58	U = 112.0, *p* = 0.98, r = 0.00
Verbal fluency	4.57 ± 7.13	2.33 ± 7.26	U = 336.5, *p* = 0.22, r = 0.20
Upper half	1.20 ± 6.30	4.53 ± 7.36	U = 81.5, *p* = 0.20, r = −0.24
Lower half	7.93 ± 6.42	0.13 ± 6.69	U = 37.5, ***p* = 0.002**, r = 0.57
category fluency	2.65 ± 4.18	1.13 ± 4.17	U = 101.5, *p* = 0.32, r = 0.18
letter fluency	6.00 ± 3.41	−0.07 ± 5.31	U = 44.0, ***p* = 0.002**, r = 0.56
Attention	−1.47 ± 7.37	0.33 ± 4.97	U = 308.5, *p* = 0.30, r = −0.13
Upper half	−3.67 ± 9.57	−1.20 ± 3.95	U = 100.0, *p* = 0.60, r = −0.10
Lower half	0.73 ± 3.24	1.87 ± 5.53	U = 80.5, *p* = 0.18, r = −0.24
Executive function	1.10 ± 2.52	0.33 ± 1.81	U = 397.0, *p* = 0.43, r = 0.10
Upper half	−0.07 ± 1.49	0.40 ± 1.55	U = 88.5, *p* = 0.30, r = −0.19
Lower half	2.27 ± 2.84	0.27 ± 2.09	U = 68.0, *p* = 0.062, r = 0.34

Values are represented as mean ± standard deviation. Significant *p*-value are shown in bold cases. Median values for verbal memory, working memory, motor speed, verbal fluency, attention, and executive function were 53.5, 21.5, 84.0, 52.8, 62.8, and 18.3, respectively. BACS: Brief Assessment of Cognition in Schizophrenia.

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
