# Peer review of "Effects of L-Theanine Administration on Stress-Related Symptoms and Cognitive Functions in Healthy Adults: A Randomized Controlled Trial"

_nutrients, 2019, doi:10.3390/nu11102362_

Round 1

Reviewer 1 Report

This is an interesting study but needs more clarification/justification on certain aspects of the clinical trial in particular.

GENERAL Comments:

The term 'non-clinical' is incorrect. It should be replaced by either 'healthy' or 'asymptomatic' to better represent the study population. A 4-week long study is not subchronic. Please remove this term throughout the manuscript and either use 'chronic' or mention the duration (4 weeks) instead. The source of L-theanine is mentioned but it is not clear if this was a synthetic product or actually isolated from green tea. This is pertinent information that should be mentioned.

CLINICAL TRIAL Comments:

How did the authors decide on enrolling n=30 subjects? Was there any statistical justification made for choosing this number of subjects? More details of the randomization process are needed. For example, what program was used to generate the sequence for randomization? What kind of safeguards were taken to maintain confidentiality and so on. Since this is a crossover study, it is unclear why treatment group had n=16 and placebo had n=14. Please explain the reasons for this discrepancy and how it could affect the results and their interpretation. Since it is mentioned as a double-blind, please describe who was blinded (in addition to study subjects). How did the authors decide on a 2-week 'washout' period before crossing over the groups? Please provide relevant references and justification. Why were subjects from such a wide age range (20-69 yrs) and both sexes chosen? It would be better to have a tighter age range and subjects from one sex for a preliminary study like this (to minimize confounding factors).

Author Response

This is an interesting study but needs more clarification/justification on certain aspects of the clinical trial in particular.

GENERAL Comments:

Comment 1: The term 'non-clinical' is incorrect. It should be replaced by either 'healthy' or 'asymptomatic' to better represent the study population.

Response 1: Thank you for the comment. We have replaced the term 'non-clinical' to 'healthy' in the manuscript (L3, 18, 54, 64, 222, 223, 228, and 286).

Comment 2: A 4-week long study is not subchronic. Please remove this term throughout the manuscript and either use 'chronic' or mention the duration (4 weeks) instead.

Response 2: Thank you for the comment. We have revised the term 'subchronic' to '4 weeks' in the manuscript (L17, 60, 63, 162, 175, 182, 201, 209, 217-218, 221, 224, 227, 231-232, 237, 242, 244, 245, 254, 256, 280, 288, 289, and Tables 2-4 and S1-2). ‘subchronic’’ of L2 was deleted.

Comment 3: The source of L-theanine is mentioned but it is not clear if this was a synthetic product or actually isolated from green tea. This is pertinent information that should be mentioned.

Response 3: Thank you for the comment. We have added the term ‘An enzymatic synthetic product’ in the methods (L90).

CLINICAL TRIAL Comments:

Comment 4: How did the authors decide on enrolling n=30 subjects? Was there any statistical justification made for choosing this number of subjects?

Response 4: Thank you for the comment. We had decided the number according to the effect sizes from our previous study in patients with MDD (Hidese et al. 2017); however, as a result, n=30 might have been rather small for healthy participants. We have revised the methods and limitation as follows: ‘The sample size was determined by using G*Power software (3.1 version) [41,42]. The current sample (30 L-theanine and 30 placebo) had a power of > 0.80, based on the effect size (0.61) of L-theanine on depression score in our previous study [40].’ (L72-74) and ‘although the sample size had been determined according to the effect size from our previous study in patients with MDD [40].’ (L272-273).

Comment 5: More details of the randomization process are needed. For example, what program was used to generate the sequence for randomization?

Response 5: Thank you for the comment. We have used a table of random numbers; therefore, we have added the phrase ‘to generate the sequence for randomization’ in the methods (L97).

Comment 6: What kind of safeguards were taken to maintain confidentiality and so on?

Response 6: Thank you for the comment. We have added the sentence ‘Confidentiality of personal information was consistently preserved by anonymizing data obtained by using code corresponding to personal identification information.’ in the methods (L82-83).

Comment 7: Since this is a crossover study, it is unclear why treatment group had n=16 and placebo had n=14. Please explain the reasons for this discrepancy and how it could affect the results and their interpretation.

Response 7: Thank you for the comment. As a result of randomization, participants were initially divided into treatment (n = 16) and placebo (n = 14) administration groups. Thereafter, they were divided into placebo (n = 16) and treatment (n = 14) administration groups, respectively. So, all 30 participants underwent total of 4 weeks treatment and placebo administration, although the timing received was not adjusted to exactly half (n =15). We, therefore, suppose that the discrepancy had little affected the results and their interpretation.

Comment 8: Since it is mentioned as a double-blind, please describe who was blinded (in addition to study subjects).

Response 8: Thank you for the comment. We have described ‘the identity of the active or placebo tablets was not disclosed to participants or research conductors at the National Center of Neurology and Psychiatry’ in the methods (L103-104).

Comment 9: How did the authors decide on a 2-week 'washout' period before crossing over the groups? Please provide relevant references and justification.

Response 9: Thank you for the comment. We have revised the method ‘After a two-week wash-out period, which was justified based on the half-life of L-theanine plasma concentration reported around 65 min in humans [50],’ citing the paper by Van der Pijl PC et al. 2010 (L98-99).

Comment 10: Why were subjects from such a wide age range (20-69 yrs.) and both sexes chosen? It would be better to have a tighter age range and subjects from one sex for a preliminary study like this (to minimize confounding factors).

Response 10: Thank you for the comment. As you point out, it might have been better to establish our inclusion criteria for a tighter age range and unisex population to minimize confounding factors. We have added the following limitation: ‘Finally, one might suspect that tighter age range and one sex population may have been more appropriate. However, we had no solid information on any specific age range or sex in which L-theanine is effective. We therefore included subjects of a relatively wider range of age and both sexes. Further studies are warranted to confirm the presented effects in much larger sample size.’ (L273-277).

Reviewer 2 Report

General comment: The research article entitled “Effects of subchronic L-theanine administration on stress-related symptoms and cognitive functions in 3 non-clinical adults: A randomized controlled trial” presents a randomized clinical study about the effect of L-thianine on mental health. This is a well-organized study, with sufficient methodology and adequate description of the results. Some minor corrections are required for the improvement of the manuscript.

Abstract: The Abstract is well written and adequately presents the aim and the basic results of the study.

Introduction: The introduction section is well-written and adequately covers the recent research about the role of L-theanine on the mental health and the importance of the present study.

Materials and Methods:  The materials and methods are adequately presented. 

-Line 71. Did authors use a specific program to calculate the sample size? Did the participants recruited to the study randomly? Please identify.

-Line 99. Did authors record the nutritional habits of the participants during the study e.g. did the participants avoid foods rich in L-theanine e.g. green tea during the study?

Results: The results of the study are analytically presented. Tables and Figures are adequate explain the findings of the study.

-Line 145-147. Authors could refer this methodological aspect in methods session e.g. 2.2. In methods session is better to be clear that this was a cross over study.

-Tables 2 and 3. The tables should be improved in order to be more readable, because they are very small.

Discussion: The results of study are sufficiently discussed.

Conclusion: The conclusion is adequate and summarizes the main text.

Bibliography/References: The references used by the authors cover adequately the relative scientific field and the aims of the study.

Author Response

General comment: The research article entitled “Effects of subchronic L-theanine administration on stress-related symptoms and cognitive functions in 3 non-clinical adults: A randomized controlled trial” presents a randomized clinical study about the effect of L-theanine on mental health. This is a well-organized study, with sufficient methodology and adequate description of the results. Some minor corrections are required for the improvement of the manuscript.

Abstract: The Abstract is well written and adequately presents the aim and the basic results of the study.

Introduction: The introduction section is well-written and adequately covers the recent research about the role of L-theanine on the mental health and the importance of the present study.

Materials and Methods: The materials and methods are adequately presented.

Comment 1: -Line 71. Did authors use a specific program to calculate the sample size? Did the participants recruit to the study randomly? Please identify.

Response 1: Thank you for the comment. Since we have used the effect sizes to calculate the sample size with ‘G*Power software’ based on our previous study (Hidese et al. 2017), we have described as follows ‘The sample size was determined by using G*Power software (3.1 version) [41,42]. The current sample (30 L-theanine and 30 placebo) had a power of > 0.80, based on the effect size (0.61) of L-theanine on depression score in our previous study [40].’ (L72-74). As participants were recruited randomly, we have revised as follows ‘All participants were recruited randomly through prepared advertisements in the local community and announcements in a free magazine or on our website.’ (L79-81).

Comment 2: -Line 99. Did authors record the nutritional habits of the participants during the study e.g. did the participants avoid foods rich in L-theanine e.g. green tea during the study?

Response 2: Thank you for the comment. Because we did not record the nutritional habits, we have described as follows: ‘Participants were instructed not to alter their usual intake of green tea or other beverage contained L-theanine during the intervention period’ (L100-102).

Results: The results of the study are analytically presented. Tables and Figures are adequate explain the findings of the study.

Comment 3: -Line 145-147. Authors could refer this methodological aspect in methods session e.g. 2.2. In methods session is better to be clear that this was a cross over study.

Response 3: Thank you for the comment. According to your comment, we have moved the part ‘After removing anonymity, L-theanine and placebo were found to have been administered to 16 and 14 participants, respectively, between weeks zero and four. Following the two-week wash-out period, subjects were administered tablets in a crossover manner between weeks six and ten.’ to the methods section (L107-110).

Comment 4: -Tables 2 and 3. The tables should be improved in order to be more readable, because they are very small.

Response 4: Thank you for the comment. We have enlarged Tables 2 and 3 accordingly.

Discussion: The results of study are sufficiently discussed.

Conclusion: The conclusion is adequate and summarizes the main text.

Bibliography/References: The references used by the authors cover adequately the relative scientific field and the aims of the study.

Reviewer 3 Report

The manuscript entitled “Effects of subchronic L-theanine administration on stress-related symptoms and cognitive functions in non-clinical adults: A randomized controlled trial” .

This is an interesting experiment concerning the effect of subchronic L-theanine administration on mental health in adults. However, the publication not include Inclusion and exclusion criteria for the study. There is no information about drug intake, diet, intake nutrients, physical activity, environmental factors - e.g. type of work. The study was conducted on a small population.

Author Response

This is an interesting experiment concerning the effect of subchronic L-theanine administration on mental health in adults.

Comment 1: However, the publication not include Inclusion and exclusion criteria for the study.

Response 1: Thank you for the comment. We have described as follows ‘Individuals who didn't have any psychiatric disorders were included, while those who had severe physical illnesses or pregnant women were excluded.’ in the methods (L77-79).

Comment 2: There is no information about drug intake, diet, intake nutrients, physical activity, environmental factors - e.g. type of work.

Response 2: Thank you for the comment. To our regret, we have not obtained such information. Therefore, we have described as follows ‘We obtained the information about current drinking alcohol and smoking habits.’ in the methods (L78-79) and ‘First, this trial was conducted in a real-world setting and possible biases (e.g., green or other kind of tea consumption, medicine intake, diet, intake nutrients, and type of work) were not accounted for,’ in the limitations (L262-264).

Comment 3: The study was conducted on a small population.

Response 3: Thank you for the comment. We have added the following limitation ‘In this context, the study was conducted on a small population and the number of the participants (n = 30) may have been insufficient, which is subject to type II errors’ (L270-272).

Reviewer 4 Report

This is a small pilot study of 30 participants.  The results of the robust double-blind, crossover, placebo controlled study show some evidence of significant beneficial effects of L-theanine.

Within the methods section the force of centrifugation is expressed in rpm, where actually the force generated is dependent on the radius of the centrifuge and the speed.  The correct unit of centrifugal force is g.

The results are clearly displayed although will be difficult to read in a printed version.  The sizes of table should be increased.  For the changes in symptom scores and cognitive function scores, significant changes are seen in only about 20-30% of measures, suggesting that the changes, although statistically significant may have little effect on daily function of the participants.  This could be emphasised in the discussion.  

Author Response

This is a small pilot study of 30 participants. The results of the robust double-blind, crossover, placebo-controlled study show some evidence of significant beneficial effects of L-theanine.

Comment 1: Within the methods section the force of centrifugation is expressed in rpm, where actually the force generated is dependent on the radius of the centrifuge and the speed. The correct unit of centrifugal force is g.

Response 1: Thank you for the comment. We have corrected the methods as follows ‘Serum and plasma samples were prepared by centrifuging blood for 10 minutes at 2,000 × g, while salivary supernatants were prepared by centrifuging saliva impregnated in mesh (Salimetrics, Co., Ltd.) for 15 minutes at 1,000 × g (Kubota Corp., Tabletop Centrifuge Model 4000, Tokyo, Japan).’ (L126-129).

Comment 2: The results are clearly displayed although will be difficult to read in a printed version. The sizes of table should be increased.

Response 2: Thank you for the comment. We have enlarged Tables 2, 3, and S2 accordingly.

Comment 3: For the changes in symptom scores and cognitive function scores, significant changes are seen in only about 20-30% of measures, suggesting that the changes, although statistically significant may have little effect on daily function of the participants. This could be emphasized in the discussion.

Response 3: Thank you for the comment. We have added the following limitation ‘Second, only about 20% of symptoms (the PSQI subscales) and cognitive functions (the BACS verbal fluency, especially letter fluency and executive function) scores showed significant changes after L- theanine administration compared to the placebo administration, suggesting that the effects are not large on daily function of the participants.’ (L265-268).

Round 2

Reviewer 1 Report

Regarding 'double blind' nature of the study, it should be disclosed if all the data collection and analysis was performed by 'blinded' investigators or not. It is not sufficient to state that only the study leads were blinded.

Author Response

Comment 1: Regarding 'double blind' nature of the study, it should be disclosed if all the data collection and analysis was performed by 'blinded' investigators or not. It is not sufficient to state that only the study leads were blinded.

Response 1: Thank you for the comment. We have added the sentence ‘All the data collection and analysis were thus performed by blinded investigators.’ in the methods (L107-108).

Reviewer 3 Report

The authors have improved the publication according to my suggestions

Author Response

Thank you for the comment.